# A New Mechanism in THRA Resistance: The First Disease-Associated Variant Leading to an Increased Inhibitory Function of THRA2

**DOI:** 10.3390/ijms22105338

**Published:** 2021-05-19

**Authors:** Sarah Paisdzior, Ellen Knierim, Gunnar Kleinau, Heike Biebermann, Heiko Krude, Rachel Straussberg, Markus Schuelke

**Affiliations:** 1Institute of Experimental Pediatric Endocrinology, Charité–Universitätsmedizin Berlin, corporate member of Freie Universität Berlin and Humboldt-Universität zu Berlin, D-13353 Berlin, Germany; sarah.paisdzior@charite.de (S.P.); heike.biebermann@charite.de (H.B.); heiko.krude@charite.de (H.K.); 2NeuroCure Cluster of Excellence; Charité–Universitätsmedizin Berlin, corporate member of Freie Universität Berlin and Humboldt-Universität zu Berlin, D-10117 Berlin, Germany; ellen.knierim@charite.de; 3Department of Neuropediatrics, Charité–Universitätsmedizin Berlin, corporate member of Freie Universität Berlin and Humboldt-Universität zu Berlin, D-13353 Berlin, Germany; 4Institute of Medical Physics and Biophysics, Group Protein X-ray Crystallography and Signal Transduction, D-10117 Berlin, Germany; gunnar.kleinau@charite.de; 5Schneider Children’s Medical Center, Petach Tikva, Israel, Department of Child Neurology, Neurogenetic Service, affiliated to Sackler School of Medicine, Tel Aviv University, Ramat Aviv IL-69978, Israel

**Keywords:** thyroid hormone receptor alpha, resistance to thyroid hormones, gain-of-function, gain-of-antagonistic function

## Abstract

The nuclear thyroid hormone receptors (THRs) are key mediators of thyroid hormone function on the cellular level *via* modulation of gene expression. Two different genes encode THRs (*THRA* and *THRB*), and are pleiotropically involved in development, metabolism, and growth. The THRA1 and THRA2 isoforms, which result from alternative splicing of *THRA*, differ in their C-terminal ligand-binding domain (LBD). Most published disease-associated *THRA* variants are located in the LBD of THRA1 and impede triiodothyronine (T3) binding. This keeps the nuclear receptor in an inactive state and inhibits target gene expression. Here, we investigated a new dominant *THRA* variant (chr17:g.38,241,010A > G, GRCh37.13 | c.518A > G, NM_199334 | p.(E173G), NP_955366), which is located between the DNA- and ligand-binding domains and affects both splicing isoforms. Patients presented partially with hypothyroid (intellectual disability, motor developmental delay, brain atrophy, and constipation) and partially with hyperthyroid symptoms (tachycardia and behavioral abnormalities) to varying degrees. Functional characterization of THRA1p.(E173G) by reporter gene assays revealed increased transcriptional activity in contrast to THRA1(WT), unexpectedly revealing the first gain-of-function mutation found in THRA1. The THRA2 isoform does not bind T3 and antagonizes THRA1 action. Introduction of p.(E173G) into THRA2 increased its inhibitory effect on THRA1, which helps to explain the hypothyroid symptoms seen in our patients. We used protein structure models to investigate possible underlying pathomechanisms of this variant with a gain-of-antagonistic function and suggest that the p.(E173G) variant may have an influence on the dimerization domain of the nuclear receptor.

## 1. Introduction

Thyroid hormone receptors (THRs) act as ligand-dependent transcription factors and positively or negatively regulate a plethora of target genes [1]. There are two genes encoding THRs (*THRA* and *THRB*). Both regulate the expression of genes which modulate development, metabolism, and growth, as well as cardiac and gastrointestinal function. The central control of thyroid function is exclusively mediated by THRB. THRs are usually organized either as homo- and hetero-dimers with their THRA/B counterpart or with retinoid X receptors (RXR). They bind to THR responsive elements (TREs) within the promoter regions of their target genes [2]. The DNA-binding domain (DBD), a conserved domain located in the N-terminus of the protein, binds to either side of the two hexameric half sites of a TRE. In their unliganded conformation, DNA-bound THRs mainly suppress their target gene expression. The suppressive conformation is relieved once triiodothyronine (T3) binds to the ligand-binding domain (LBD) of the THRs, thus clearing the way for target gene transcription [3]. T3 is the active form of thyroid hormone, which is generated in many cells by deiodination of the prohormone thyroxine (T4). Thus, in the absence of T3, as is the case in severe hypothyroidism, the T3 responsive THRs are locked in their suppressive conformation, thereby causing symptoms of hypothyroidism, such as developmental delay, intellectual disability, hypometabolism, short stature, bradycardia, and constipation.

In *THRA*, mRNA-splicing at an alternative splice site in exon 9 results in a shorter isoform lacking the amino acids (AA) encoded by exon 10 (Appendix A). This isoform of 410 AA (THRA1) can bind two molecules of T3. A second isoform of 490 AA with inclusion of amino acids encoded by exon 10 (THRA2) has an elongated C-terminus but cannot bind T3 [3]. Since THRA2 is unable to bind to T3 [4] but can still bind to DNA and form heterodimers with THRA1, it has a dominant negative effect on THRA1 transactivation activity [5,6]. Since the expression of THRA and THRB as well as their splicing isoforms (THRA1 and THRA2) are highly regulated in a cell- and time-specific manner [7,8,9], the actual target cell response to T3 can be considered as the sum of the individual effects of the interplay between all THR isoforms. At present, a detailed description of such THR interplay in the development and function of target cells and tissues is still incomplete.

Patients with disease-causing *THRB* variants were already described in the 1980s, because they presented with increased T3 and T4 blood concentrations [10]. Due to the involvement of THRB in the central regulation of thyroid function *via* the hypothalamus–pituitary–thyroid axis (HPT), the negative feedback loop regulated by TH is disturbed in THRB resistance. Since the *THRA* gene is intact in these patients, the elevated T3 levels result in hyperthyroid symptoms, especially in target tissues which preferentially express *THRA* (e.g., heart and brain), leading to hyperactivity and tachycardia [11,12], while some patients may also be paucisymptomatic.

In contrast to THRB resistance, the first pathogenic variants in *THRA* have only been published rather recently [13]. This was probably due to the fact that disease-causing *THRA* variants, contrary to *THRB*, do not affect central TH regulation. Only after the advent of whole exome sequencing was a first disease-causing *THRA* variant discovered in a child whose symptoms were suggestive of hypothyroidism, but who had normal TH-blood levels [13].

Since then, 29 different disease-causing *THRA* variants have been found and investigated (Appendix A). Most of those patients suffer from growth retardation, mild-to-moderate intellectual disability, a low basal metabolic rate, mild skeletal dysplasia with specific facial features, normocytic anemia, bradycardia, and severe constipation. These symptoms reflect the expression pattern of THRAs, which are found predominantly in the central nervous system, gastrointestinal tract, heart and muscle tissue, as well as in the bones [14,15,16,17,18,19,20]. TSH levels are generally normal. T4 levels are below or in the low–normal range, while T3 levels in contrast are in the high–normal or above the reference range.

Most of the published disease-causing THRA variants are located in the LBD close to the C-terminus, which is encoded by exons 8 and 9 (Appendix A and Appendix A). These THRA variants lead to loss of T3-binding capability. Such variants change the hormone responsive THRA1 isoform into a hormone unresponsive conformation that suppresses target gene expression in a similar way as the splice variant THRA2. As a consequence, such dominant negative variants are inherited as a dominant Mendelian trait. So far, 16 missense variants located in exons 1–8 and part of exon 9 are shared between THRA1 and THRA2. Most of these variants were confirmed to disturb T3-binding of THRA1 with the same dominant negative effect as described for the exon 9 variants [14,15,17] (Appendix A).

Here, we describe a new disease-causing *THRA* variant in exon 6, which is present in both THRA1 and THRA2 isoforms. Surprisingly, our molecular analysis revealed that this new variant does not disrupt T3-binding of THRA1. This finding stands in stark contrast to all disease-causing *THRA* variants known so far (see Appendix A for all known *THRA* variants published to date). On the contrary, we show that the T3-response of the THRA1 variant is even increased. At the same time, the variant also enhances the suppressive effect of THRA2 on the transcriptional activation of THRA1. We suppose that the effect of this p.(E173G) variant is based on a new mechanism related to the modification of the functionally relevant interplay between THRA1 and THRA2, which in consequence causes clinical symptoms that in part mimic “classic” features of patients with previously reported *THRA* variants.

## 2. Results

### 2.1. Case Reports

Our patients are two siblings born to non-consanguineous Jewish parents. All research-related investigations were performed after recording written informed consent according to the Declaration of Helsinki, including permission for the publication of full facial photographs. The clinical features of the patients are listed in Table 1.

#### 2.1.1. Patient II:4

The now 18-year-old patient (Figure 1A) is the fourth child of her parents. She was born at 41 weeks of gestation with a normal birth weight of 3,185 g (–1.23 z). Congenital equinovarus foot deformities were corrected surgically and with orthopedic devices. Her motor development was delayed (independent sitting at 14 months, free walking at 24 months, slow initiation of movements) and her gait remains ataxic until present. Her speech development was slow, with first words spoken at 2.5 years and dysarthric pronunciation from then on. Tooth eruption was delayed by 7 months. Cognitive assessment at 7 years confirmed mild intellectual disability with an intelligence quotient (IQ) of 60. Behavioral problems became apparent at 5 years with mainly aggressive and hyperactive behavior. Since age 7, she has been on high-dosage methylphenidate (1.5 mg/kg body weight) to treat her ADHD. Obesity became apparent at 12 months and Prader–Willi syndrome was ruled out. From early childhood onwards, the patient suffered from severe constipation, requiring regular enemas for defecation. Kyphoscoliosis progressed from a Cobb angle of 31° at 12 years to 42° at present. Due to severe headaches and developmental delay, a cranial MRI was repeated at 8 years of age. The MRI showed normal results apart from slightly widened sulci. Her height is presently at 155 cm (–2.05 z). However, due to her scoliosis, we calculated 0.9 cm relative loss of body height. The corrected height of 156 cm would place her at –1.89 z and her BMI at 44.4 kg/m^2^ (+4.12 z). Her menarche was normal at 12 years, and at 13 years, serum IGF-1 values and bone maturation were age appropriate. Bone alterations as described in other patients carrying disease-causing THRA variants were absent on X-ray of the limbs and on cranial MRI. We did not see anemia or skin abnormalities. Her heart rate was tachycardic (>100/min) and never bradycardic.

#### 2.1.2. Patient II:2

The now 29-year-old elder sister (Figure 1A) of patient II:4 is the second child of her parents. Her birth weight was 3,916 g (+0.66 z) after 37 weeks of gestation. Obesity was observed from the second year of life and Prader–Willi syndrome was ruled out. She also suffered from severe constipation from early childhood onward. She did not speak until 2 years of age. She appeared to suffer from intellectual disability, and objective assessment at the age of 8 years confirmed borderline intellectual disability with an IQ of 68. Her movements were slow. Cranial MRI at 17 years of age showed general brain atrophy with widening of the sulci. A repeated cranial MRI at 20 years confirmed loss of brain volume with widening of the sulci and normal inner ventricles. Due to behavioral disturbances with ADHD, she has been treated with methylphenidate (0.3 mg/kg body weight) since 10 years of age. At present, her height is 161.6 cm (–1.02 z) and her body weight 115.6 kg (+3.88 z; BMI = 44.3 kg/m^2^, +4.11 z).

### 2.2. Mutation Screening

In the WES dataset we discovered a novel heterozygous missense variant in *THRA* (chr17:g.38,241,010A > G, GRCh37.13 | c.518A > G, NM_199334 | p.(E173G), NP_955366) in both patients II:4 and II:2. The variant exchanges an evolutionarily highly conserved glutamine for a glycine (Figure 1B). As the parents were unrelated, we had first applied a filter model, assuming dominant inheritance. The dominant model yielded 10 candidate variants that caused an amino acid exchange, at the same time having a low or zero minor allele frequency in the gnomAD database. Subsequently, only the *THRA* variant was shortlisted due to the cellular functions of this gene and the previously described phenotype. Both healthy siblings and the mother show the WT sequence at the variant’s position. The patients’ father, who is significantly obese and also suffers from constipation and shows significant macrocephaly, also harbors the missense variant, which is not listed in the gnomAD variant database. The here described *THRA* variant (ClinVar VCV000992628) fulfills the PM2 criteria of the American College of Medical Genetics and Genomics classification [21], placing it in the ‘likely pathogenic’ category. cDNA studies on RNA extracts from patient fibroblast cell lines confirmed transcription of variant and WT alleles in nearly equimolar amounts for both *THRA1* and *THRA2* splicing isoforms (Figure 1B). Relative abundance of mRNA transcription was lower for the *THRA2* isoform (Figure 1D), as expected for non-neuronal tissue [9].

### 2.3. Functional Characterization of the Novel THRA Variant.

As the patients’ phenotype was compatible with the previously described symptom spectrum of THRA-resistance, we tested the THRAp.(E173G) variant for its in vitro effect on THRA function (Figure 2). We used a TRE-containing luciferase reporter to test the T3 inducible transcriptional activity of the THRAp.(E173G) variant in comparison to WT protein and a previously published THRAp.(A263V) partial loss-of-function variant located in exon 8 [17]. This allowed investigation of the new THRAp.(E173G) variant’s impact on T3-inducible transactivation *via* THRA1 as well as on THRA2-mediated suppression. We chose human placental choriocarcinoma cells (JEG3) as our cell model to ensure comparability to previously investigated *THRA* mutations [13,14,15,17,18,19]. The basic THR expression levels in JEG3 are not known. However, as also mock-transfected cells somewhat respond to T3 (Appendix A), we suspect low-level THR expression in JEG3 cells. This finding has been reported by other authors as well [22]. As TH action seems to be essential for cell survival, we do not know of any cell lines that entirely lack any THR expression. 

While the THRAp.(A263V) variant decreased the T3 response of THRA1, as expected [17], introduction of the p.(E173G) variant into THRA1 surprisingly did not lead to a reduction of transcriptional activity, but rather to an increase of the T3 response, however, without reaching statistical significance. Determination of the half maximal effective concentration, EC50 ± SEM [THRA(WT) ≥ 400 pM; THRAp.(E173G) ≥ 943 ± 2.8 pM] suggests no relevant change in affinity of T3 to THRA1p.(E173G) in comparison to the loss-of-function variant THRA1p.(A263V), with a significantly lower T3 affinity in the nanomolar range [103 ± 1.7 nM] (Figure 2A).

To mimic the physiological state of heterozygosity, we next transfected constructs for either THRA1(WT) alone or in combination with THRA1p.(E173G). Here, the THRA1p.(E173G) variant enhanced the T3 response with a clear gain-of-function effect but again without change of T3 affinity [THRA1(WT) + THRA1(WT) ≥ 340 ± 1.7 pM, THRA1(WT) + THRA1p.(E173G) ≥ 113 ± 3.6 pM] (Figure 2B). This again stands in stark contrast to all so-far published disease-associated THRA variants which all exert a dominant negative effect through loss-of-T3-function if co-transfected with the WT construct [23,24].

As in heterozygous patients also mutant:mutant THRA1 homodimers do occur, albeit at a lower percentage of only 25%, we tested the effect of exclusive THRA1p.(E173G) homodimers on T3-dependent transactivation by doubling the amount of transfected DNA. We observed a massive stimulation of the reporter, suggesting a super-additive gain-of-function effect of the variant, which appears to be gene-dosage-dependent. Again, there was no significant effect on T3 affinity [THRA1(WT) + THRA1(WT) ≥ 209 ± 2.3 pM, THRA1p.(E173G) + THRA1p.(E173G) ≥ 330 ± 3.0 pM] (Figure 2C).

The p.(E173G) variant is encoded by exon 6 sequences and thus shared between both THRA splice isoforms (THRA1 and THRA2). These isoforms differ both in molecular weight and in their C-termini, which are encoded by exons 9 and 10 (Figure 3B, Appendix A). Due to these differences and the resulting change in its LBD, THRA2 is unable to bind T3, but is still able to dimerize with THRA1. These heterodimers antagonize T3-induced THRA1 function, especially if cellular THRA2 levels are higher than THRA1 levels. We thus tested the effect of the p.(E173G) variant on THRA2 function by co-transfection of THRA2p.(E173G) and THRA1(WT) constructs (Figure 2D,E). While equimolar transfection of THRA2(WT) did not suppress the T3-induced THRA1(WT) function, as reported previously [6], an equimolar amount of mutant THRA2p.(E173G) significantly reduced T3-induced THRA1 activity (Figure 2D,E). As an increase of the THRA2:THRA1 ratio would progressively suppress THRA1 T3-mediated transactivation activity, we tested such an increase in the THRA2(WT) and THRA2p.(E173G) mutant system. In this experiment, transactivation activity for a THRA1(WT):THRA2p.(E173G) ratio of 1:1 was even significantly lower than a THRA1(WT):THRA2(WT) ratio of 1:5, thereby confirming the increased inhibitory effect of THRA2p.(E173G) (Figure 2D). A similar result was observed in a T3-titration curve featuring equimolar 1:1 mixtures between the different WT and mutant isoforms (Figure 2E). In conclusion, the p.(E173G) variant rendered THRA2 into a more potent antagonist of THRA1 function.

### 2.4. Insight from THR 3D Structures and Homology Models

The p.(E173G) substitution is located outside of the T3-binding pocket(s) and is oriented towards the surface of the protein (Figure 3A). Therefore, we suppose that in contrast to other disease-associated THRA variants, e.g. THRAp.(A263V) and THRAp.(R384C), the underlying molecular mechanism of the THRAp.(E173G) variant does not interfere with the T3-binding process.

Further, it is unlikely that the p.(E173G) variant has a direct impact on the THR:DNA interaction, as Glu173 is located outside of the DNA-binding domain. However, an indirect link through changes in the helical structure (helix 1 of the LBD) around Glu173 and the hinge region in close proximity cannot be excluded.

Homodimer structures of THRB and other nuclear receptors suggest the possibility of different homodimer interfaces (diverse protomer arrangements) between the protomers, as depicted in Figure 3A (Glu173 is located outside of the dimer-interface) in contrast to Figure 3C (Glu173 should be inside of the dimeric interface). Presently, we do not know which dimer interface is functionally relevant, especially during *in vivo* interaction between the two THRA splicing isoforms or in case of the pathogenic condition (mutated THRA variants).

### 2.5. Co-Expression of WT and Variant THRA1 with the Retinoid X Receptor α (RXRα)

RXRα is a further protein that physiologically dimerizes with THRA1 [2] and inhibits THRA1-mediated transactivation [24]; hence, we tested the influence of the p.(E173G) variant on this inhibition with the TRE-containing luciferase reporter. We prepared a series of construct mixtures each containing the same concentration of either *THRA1*(WT) or *THRA1*p.(E173G) construct, adding decreasing amounts of *RXRα* constructs in a dilution series from 1:1 down to 1:0.1 (Figure 2F). In order to keep the total DNA concentration identical for each transfection, the wells with falling RXRα concentrations were compensated for with empty vector. The addition of RXRα inhibited THRA1(WT)-mediated transactivation as expected. However, we did not see any statistically significant difference between *THRA1*(WT) and *THRA1*p.(E173G) genotypes (Figure 2F). This let us conclude that the interaction between THRA1 and RXRα was unaffected by the p.(E173G) variant. This is in agreement with insight from a solved THRB:RXRα heterodimer structure (Figure 3D) that suggests that Glu173 is not part of the RXRα:THRA dimerization interface (Figure 3D).

## 3. Discussion

Disease-causing variants in *THRA* have only been diagnosed after the advent of whole exome sequencing (WES) [13]. Prior to this more ‘unbiased’ and less phenotype-centered approach, the apparently normal thyroid hormone serum levels in these patients prevented an earlier diagnosis of this particular thyroid dysfunction, despite the fact that clinical key symptoms of hypothyroidism, such as developmental delay, constipation, obesity, and bone/skin symptoms, were present. At first sight, the symptoms of our patients were suggestive of hypothyroidism, but nonetheless did not lead to the diagnosis of thyroid hormone dysfunction due to their normal TSH, T4, and T3 serum concentrations. Again, it was only WES that revealed this disease-causing variant in *THRA*.

The p.(E173G) variant is of particular interest because (i) of all reported genetic alterations in THRA (Appendix A), it is among the very few located between the DBD and LBD [23], (ii) it is encoded by exon 6 and hence shared by both splicing isoforms [4,5], (iii) it was not described before, and (vi) the patients exhibited an uncommon “compound” phenotype with tachycardia and behavioral problems necessitating treatment with methylphenidate.

We thus investigated the functional effect on THRA1p.(E173G) and THRA2p.(E173G). Unexpectedly, the p.(E173G) mutant did not abrogate T3-induced THRA1 function, but, on the contrary, led to its enhancement. This finding stands in stark contrast to all other so-far investigated disease-associated THRA variants, including the p.(A263V) variant that we tested in parallel in our experimental system [17]. Such a gain-of-function effect was also shown for the THRA1(WT):THRA1p.(E173G) heterodimer, which had a dominant negative effect in all other disease-associated THRA variants described to date.

As mentioned before, THRA2 is unable to bind T3, but can still form dimers with THRA1 and bind DNA, thereby inhibiting THRA1 transactivation. Therefore, we tested the impact of the p.(E173G) variant on this THRA2-mediated inhibition of transactivation. We discovered that THRA2p.(E173G) had an increased inhibitory effect on THRA1(WT) because the T3 response at 10 nM was curtailed by 50% after replacement of THRA2(WT) by THRA2p.(E173G). The amount of THRA2(WT) construct had to be increased by a factor of ten to reach comparable inhibitory activity (Figure 2D).

This unexpected gain-of-(inhibitory)-function of THRA2p.(E173G) motivated us to study the potential underlying molecular backgrounds of this variant and to measure effects using structural information. First, based on the model of the 3D crystal structure of THRA1 in complex with T3, we can conclude that the mutated Glu173-residue is located outside of the T3 binding pockets(s) of the LBD (Figure 3A), which could explain the normal stimulation of THRA1p.(E173G)-mediated transactivation by T3 (Figure 2A). In order to explain both increased T3 transactivation of THRA1p.(E173G) and increased inhibitory function of THRA2p.(E173G), we further explored the spatial position of Glu173 with regard to other features typical for a nuclear receptor, such as (i) dimerization, (ii) cofactor-binding, and (iii) DNA-binding.

From the proposed structural THRA1 dimer assembly based on already determined THRB complexes (Figure 3A–D), we conclude that Glu173 is located on the surface of the receptor, outside of the DNA-binding domain. However, a potential link to the DNA-binding domain *via* the hinge region located in close proximity (adjacent to helix 1) cannot be excluded. Anyhow, the localization on the protein surface rather suggests a role for Glu173 in protein–protein interactions, e.g., either with cofactors or for homo- and hetero-dimerization. The removal of the large and negatively charged side chain of glutamic acid in the p.(E173G) mutant could in principle impact any intermolecular interaction of THRA. Dimerization could be enhanced by forcing a new dimer interface (Figure 3C) that lacks the large negatively charged side chain of Glu173, or an existing dimer interface might be strengthened. In any case, both THRA1(WT):THRA1p.(E173G) and THRA1(WT):THRA2p.(E173G) dimers should be more stable and thus more active. The net result of stimulation *versus* inhibition of T3-mediated transactivation in a cell would entirely depend on the molar ratios between the two splicing isoforms (Figure 4).

Finally, given the fact that Glu173 is located opposite of the THRA1:RXRα interaction interface (Figure 3D), we would expect that the replacement of glutamine by glycine does not interfere with RXRα-mediated inhibition of THRA1 transactivation. Indeed, upon T3-stimulation, RXRα inhibits THRA1(WT) and THRA1p.(E173G) transactivation to the same degree. We imagine that such opposing effects on the T3-response would also result if the p.(E173G) variant enhanced DNA-binding of THRA1p.(E173G) and THRA2p.(E173G).

The pathophysiology of all so-far reported disease-associated *THRA* variants was mediated through a loss-of-T3-function, leading to a dominant negative effect of the mutant THRA on the wild-type THRA in heterodimers. Thus, the variant is inherited in a dominant mode and heterozygous patients are affected [23,25]. Here, we describe a new pathomechanism for a THRA variant that does not interfere with T3-binding but rather with further receptor features such as dimerization and DNA-binding. Disease-causing variants in androgen receptors (AR), which have a high homology to the THRs, were likewise shown to interfere with ligand-binding, but also affect receptor dimerization and DNA-binding. Indeed, for some particular gain-of-function AR-variants, authors discussed an increased dimerization effect through variants that promote tumor progression of prostate cancer [26]. Comparable effects of disease-causing variants in THRA and AR are likely, as both share the common evolutionarily conserved nuclear receptor features and topologically equivalent interactions. However, such a theory needs to be further substantiated by protein–protein binding and 3D co-crystallization studies.

Irrespective of the exact molecular mechanism, as in vitro and in situ characterizations can only provide an idea of the physiological conditions, the opposing functional effects of the p.(E173G) variant, that we discovered with our reporter gene assays, imply a complex functional constellation in the cells of our patients (Figure 4). As the splice variants *THRA1* and *THRA2* show a different spatio-temporal expression [7,27], the constellations between four different transcripts from the same *THRA* gene determine the individual cellular T3-response: (i) THRA1(WT), (ii) THRA1p.(E173G), (iii) THRA2(WT), and (iv) THRA2p.(E173G) (Figure 4). In cells that predominantly express THRA1, the T3-response will be augmented with a “hyperthyroid” cellular phenotype, while in cells that mainly express THRA2, the T3-response will be suppressed, leading to a “hypothyroid” phenotype. In cells expressing both THRA1 as well as THRA2, the effects of increased THRA1p.(E173G) activation and stronger THRA2p.(E173G) suppression may cancel out each other, resulting in a euthyroid state. Very recently, the precise balance between THRA1 and THRA2 expression was described on the tissue level in the adult mouse, with the striking finding that brain and neuronal tissues predominantly express THRA2 [9]. Provided that a similar distribution would be present in humans, for which we have evidence on the transcriptional level [27], we could expect our patients with the THRAp.(E173G) variant to be mainly hypothyroid in central neurons (THRA2 > THRA1: increased inhibition) and hyperthyroid in cardiomyocytes (THRA2 < THRA1: increased transactivation) [9]. When trying to square the phenotype of our patients with these mouse data, we find a striking correspondence: the neuro-cognitive symptoms of the patients argue for a hypothyroid brain phenotype that fits well with a predominant THRA2 expression in the nervous system. In addition, a hyperthyroid over-stimulation during brain development *via* the THRA1p.(E173G) receptor variant may cause neuro-cognitive dysfunction and behavioral abnormalities as well, having been described in patients with congenital hyperthyroidism, who also suffered from intellectual disability and hyperactivity [28]. Thus, both effects of the p.(E173G) variant, the increased inhibition of THRA2p.(E173G), and the increased stimulation *via* the THRA1p.(E173G) may in principle cause defects of neurodevelopment.

In peripheral tissues, the increased stimulation *via* THRA1p.(E173G) seems to cause hyperthyroid symptoms because the resting heart rate of patient II:4 was rather high, suggesting a hyperthyroid state of the heart.

Other hypothyroid symptoms that have been described in *THRA* variant carriers (e.g., in bone, skin, and blood) were absent in our patients (Table 1), suggesting a rather euthyroid level in non-neuronal and non-cardiac cells.

In conclusion, the new gain-of-function variant in our patients results in hypothyroid as well as hyperthyroid symptoms that occur side by side depending on the specific tissue(s) and their relative THRA1 *versus* THRA2 expression levels. Depending on the cellular splicing patterns (*THRA1 versus*
*THRA2*) the same THRAp.(E173G) variant may either render an affected cell hyper- or hypo-thyroid. We need to find more patients with such “non-classical” gain-of-function *THRA* variants. The comparison of their phenotypes with those of “classical” patients with loss-of-T3-function variants will help to further delineate this distinct phenotype subset and its pathophysiology. However, we anticipate that the spectrum of phenotypes will be very broad, as even in our single family, the three variant carriers presented with a broad range of symptoms and severities. We have to take into consideration the influence of multiple factors, whose effects are integrated into a final T3-response. Such factors would comprise polymorphisms in cofactors, in DNA-response elements, and in the splicing machinery.

## 4. Materials and Methods

The study was approved by the institutional ethics review boards at Schneider Children’s Medical Center, Petah Tikva, Israel (0643-13-RMC) and Charité-Universitätsmedizin Berlin, Germany (EA2/107/14, Ethikkommission der Charité, Ethikausschuss 2 am Campus Virchow-Klinikum at 01.10.2014).

### 4.1. Genetic Analysis

DNA was isolated from peripheral leukocytes. Whole-exome sequencing (WES) was performed in both patients. Exonic sequences and flanking intronic regions were captured using the SureSelect^®^ human all exon V5 kit (Agilent Technologies, Santa Clara, CA, USA) and sequenced on an Illumina HiSeq 4000 machine yielding 25.5|28.5 Mio paired-end FASTQ reads. These were aligned to the human GRCh37.p11 genomic reference with BWA-MEM v0.7.8., and 96.3|97.4% of the SureSelect^®^ V5 positions were covered > 10×, and 92.5|93.4% > 20×. After fine-adjustment, the raw alignments were called for deviations from the reference sequence in all coding exons and 50 bp flanking regions using GATK v3.8. The resulting VCF file was analyzed with MutationDistiller, a freely available online tool for user-driven analyses of Whole Exome Sequencing data to assess potential pathogenicity of all variants [29]. Phenotypic information was included *via* the Human Phenotype Ontology (HPO) with the search terms “obesity” (HP:0001513), “constipation” (HP:0002019), and “intellectual disability” (HP:0001249). The suspected mode of inheritance was dominant (strict). Potentially disease-causing variants were visually inspected using the IGV software (v2.8.12) [30]. Variants were tested for segregation in all family members by automatic Sanger sequencing using the BigDye^®^ Terminator v3.1 chemistry (Applied Biosystems, Waltham, MA, USA) on the ABI 3500 Genetic Analyzer. For verification of the *THRA* variant and for segregation testing, we used the oligonucleotide primers FW 5′-acc ttg tgc ctc tct gtt ca-3′, REV 5′-ctc act ctc ttc tcc ctg cc-3′.

### 4.2. Cloning, Cell Lines, and Chemicals

The cDNA of both isoforms of human *THRA* (Plasmid ID HsCD00078513 and HsCD00005562) and of retinoid receptor α (*RXRα*, Plasmid ID HsCD00079702) were purchased from the DNASU plasmid repository [31] and cloned into the eukaryotic expression vector pcDNA3. The respective variants were introduced using site-directed mutagenesis. The T3-responsive luciferase-based reporter gene with a TRE consisting of four direct repeats (DR4) was a kind gift from Josef Köhrle at the Institute of Experimental Endocrinology (IEE, Charité Berlin, Germany) [32].

For in vitro testing, human placenta choriocarcinoma (JEG3) cells were cultivated in Dulbecco’s modified Eagle’s Medium (DMEM, Biochrom, Berlin, Germany) supplemented with 10% fetal bovine serum (FBS, Gibco, Carlsbad, CA, USA) and 0.5% Penicillin/Streptomycin (Gibco). Cultivation took place at 37 °C under humidified air containing 5% CO_2_. For the reporter gene assays, cells were seeded into 96-well plates (Falcon, Kaiserslautern, Germany) at 20,000 cells per well in DMEM supplemented with 2% charcoal-treated FBS (Sigma-Aldrich, Darmstadt, Germany) to ensure a TH-free environment. Triiodothyronine (T3) was purchased from Sigma-Aldrich (Darmstadt, Germany) and solved in DMSO, resulting in a stock concentration of 10 mM that was diluted to the indicated concentrations using DMEM (Gibco, Carlsbad, CA, USA) without supplements. We grew fibroblast cell lines of all three family members who carried the p.(E173G) variant (Figure 1C) from skin punch biopsy samples in DMEM (Gibco, Carlsbad, CA, USA) with 10% FBS (Gibco, Carlsbad, CA, USA) and 1% Penicillin/Streptomycin (Gibco, Carlsbad, CA, USA) and collected the cell pellets at each passage for RNA extraction.

### 4.3. Amplification and Analysis of cDNA from Fibroblasts

RNA from fibroblasts was extracted using the RNeasy Mini Kit (Qiagen, Hilden, Germany) according to the manufacturer’s protocol. Reverse transcription was carried out using reverse transcriptase M-MLV (Promega, Walldorf, Germany) and oligo-dT primers (Promega). RNA was protected with RNAsin Plus (Promega, Walldorf, Germany). Resulting cDNA was then used as a template to amplify the mRNA of both *THRA* isoforms with primers FW 5′-acc gca aac aca aca ttc cg-3′ (shared by both isoforms), REV 5′-ggg gca ctc gac ttt cat gt-3′ (*THRA1*), and REV 5′-att ccg aga agc tgc tgt cc-3′ (*THRA2*) purchased from Sigma-Aldrich (Darmstadt, Germany). Amplicons were then analyzed with gel electrophoresis and automatic Sanger sequencing using the BigDye^®^ Terminator v3.1 chemistry (Applied Biosystems, Waltham, MA, USA) on the ABI 3130 xL Genetic Analyzer.

### 4.4. Transfection

Before we started the experiments, we performed several transfection optimization experiments, following the recommendations of the manufacturer of Lipofectamine^®^ 2000 (ThermoFisher, Waltham, MA, USA). For that, a constant amount of DNA (2.5 µg per sample) was distributed in equimolar amounts between two plasmids, one containing the reporter and the other the THRA gene. We tested different cell numbers and amounts of transfection reagent according to the manufacturer’s protocol.

Based on the results obtained from these experiments, we decided upon 20,000 cells per well and a 1:1 (*v*/*v*) ratio of DNA:Lipofectamine^®^ 2000. The setup was optimized for transfection with THRA1 + empty vector + reporter, which would represent the expression of a single (wild-type *versus* mutant) allele. The total amount of transfected DNA was kept constant between the experiments. Hence, 24 h after seeding, cells were transfected with Lipofectamine^®^ 2000 in a DNA:reagent ratio of 1:1 in Opti-MEM^®^ (Gibco, Carlsbad, CA, USA). Cells were co-transfected with the DR4-reporter gene and either a stable amount of *THRA1* wild type (WT), mutant, or a combination of WT/mutant for the transactivation assays, as well as a stable amount of *THRA1(WT)* and increasing amounts of *THRA2(WT)* or *THRA2p.(E173G)* for dominant-negative effects. In all cases, the total amount of recombinant DNA was kept constant for all transfections between the assays, so empty pcDNA3 vector DNA was added up to the final amount if necessary. Transfected cells were placed back into the incubator at 37 °C and humidified air containing 5% CO_2_.

In order to model heterozygosity for the mutation as in the patients, we added respective mixtures of the *THRA(WT)* + *THRAp.(E173G)* plasmids + reporter, again with a constant amount of total DNA. We are aware that in vitro systems are always artificial and can only provide an idea of the physiological mechanisms.

### 4.5. T3 Responsive Luciferase Assay

Forty-eight hours after transfection, growth media was removed, and cells were stimulated with the indicated amount of T3 (50 µL/well) in DMEM without supplements for 24 h at 37 °C to ensure sufficient luciferase expression. Subsequently, cells were lysed using passive lysis buffer (Promega, Walldorf, Germany) at room temperature according to the manufacturer’s protocol. Cell lysates were stored at –20 °C. The measurement of luciferase activity was obtained by transfer of 10 µL cell lysate to white opaque 96-well plates, injection of luciferase substrate (Promega, Walldorf, Germany, 40 µL/well), and measurement of light emission using a plate reader (Mithras LB 940, Berthold Technologies, Bad Wildbad, Germany) (Figure 2).

### 4.6. THR Structures to Study the Putative Molecular Function of Glu173

To gain insight into the potential molecular pathomechanism related to the p.(E173G) substitution, we searched for solved structures of THRs in the PDB protein structure database: www.rcsb.org [33]. Besides the partial THRA1 monomeric structure in complex with T3 (PDB ID: 4lnw [34]) and T4 (PDB ID: 4lnx [34]), diverse THRB ligand-binding domain structures were also available in complex with thyroid hormone as homodimer (PDB ID: 3d57 [35]), as homodimer with thyroid hormone mimetics (PDB ID: 1n46 [36]), or as DBD complexed with DNA (PDB ID: 3m9e [37]). Those structures can be used to study THRA aspects under the assumption of high functional and structural homology between THR subtypes (sequence similarity on the AA level between THRA1 and THRB >75%). In addition, a THRB structure with a bound cofactor peptide (PDB ID: 1bsx [38]) and in a heteromeric constellation with RXR (PDB ID: 4zo1 [39]) were also available and provided structural information of potential protein interfaces and intermolecular interactions. These structures were used to obtain information about the potential involvement of the THRAp.(E173G) substitution in ligand:DNA-binding or intermolecular interactions (Figure 3). Specifically, putative dimer formations already determined for THRB superimposed with THRA1 monomers provided hints for dimer arrangements of TRH’s and linked dimer-interfaces. However, structural information for THRA2 and associated specificities (e.g., its extended C-terminus) are still missing. For protein superimposition and visualization of the structural complexes we used the software Pymol (PyMol Molecular Graphics System, Version 1.5; Schrödinger, LLC., New York, NY, USA), the sequence alignment presented in Figure 3 was prepared by using the software BioEdit [40].

### 4.7. Statistical Analysis

Statistical analysis was performed using GraphPad PRISM 6 (GraphPad Software Inc., La Jolla, CA, USA). The respective statistical tests are mentioned in the figure legends. Statistical significance was set at * *p* ≤ 0.05, ** *p* ≤ 0.01, and *** *p* ≤ 0.001. Data was obtained from a number of 3–7 independent experiments (indicated at each panel). For each of the repetitive experiments, cells were taken from a different cell passage and were transfected individually (in technical triplicates) with a transfection master mix containing the respective combination of DNA containing THRA and reporter gene. Concentration–response curves of each experiment were analyzed by fitting a non-linear regression model for sigmoidal response using GraphPad PRISM 6.

## Figures and Tables

**Figure 1 ijms-22-05338-f001:**
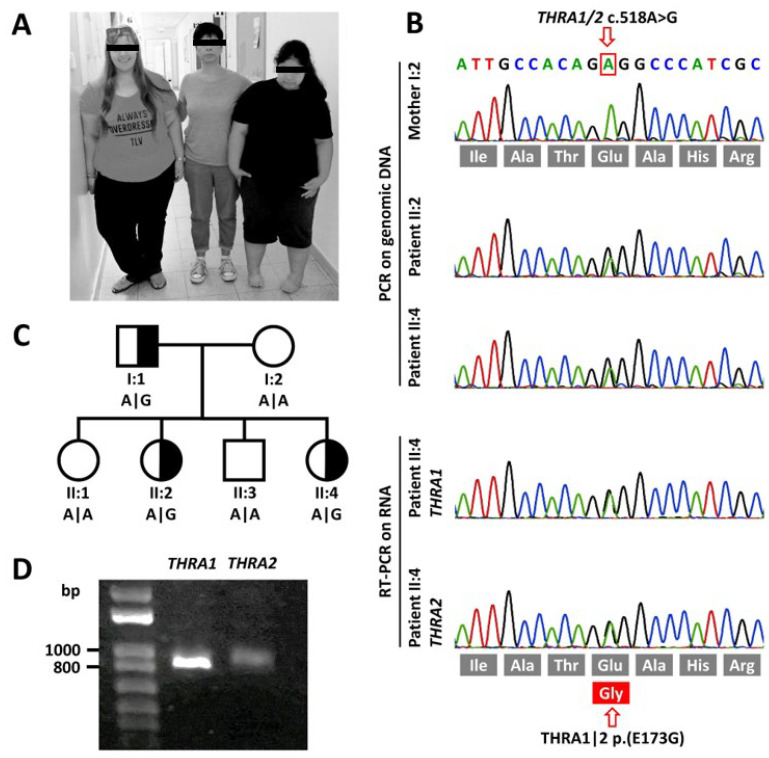
Clinical and molecular genetic findings. (**A**) Photographic image of the mother (I:2) with her affected elder (II:2, left side) and younger (II:4, right side) daughters. (**B**) Sanger sequencing traces of genomic DNA of both affected sisters and their mother depicting the heterozygous variant in *THRA*. cDNA sequencing of mRNA extracted from cultured skin fibroblasts of patient II:4 shows that both WT and variant alleles are both expressed albeit at different levels. (**C**) Pedigree of the family depicting the genotypes at cDNA position c.518. (**D**) RT-PCR of *THRA1* and *THRA2* splicing isoforms of patient II:4 shows both isoforms to be expressed. However, as expected [9], in fibroblasts, THRA2 is expressed at lower levels as THRA1.

**Figure 2 ijms-22-05338-f002:**
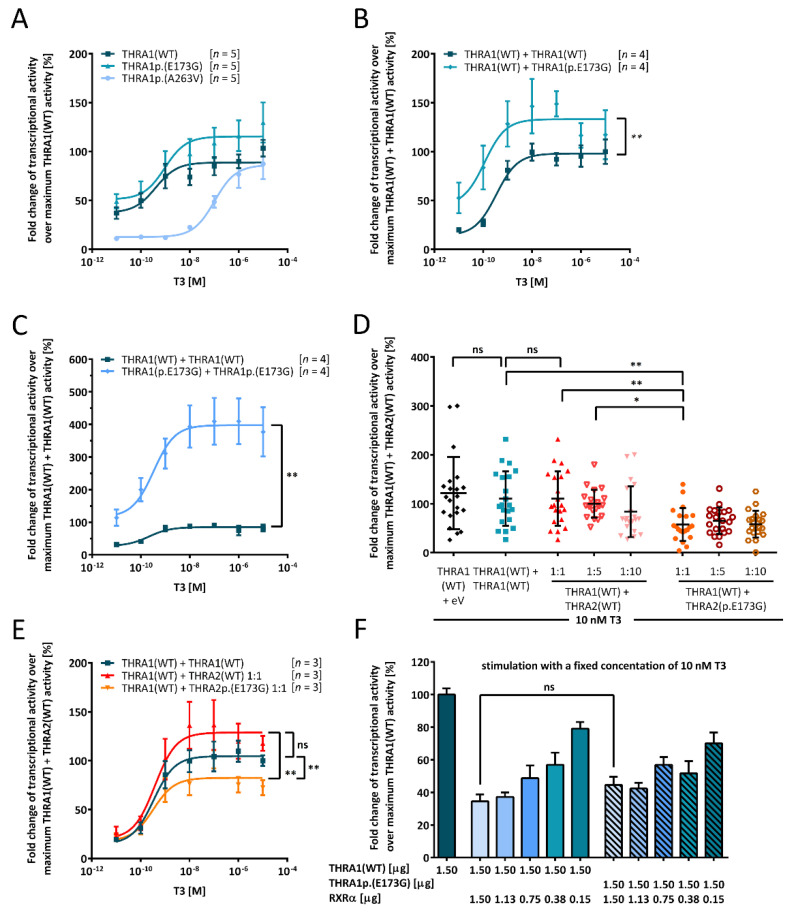
Functional in vitro characterization of the THRA1p.(E173G) variant. JEG3 cells were co-transfected with T3-responsive luciferase reporter together with the indicated THRA variants: (**A**) Comparison of THRA1(WT) (dark blue squares), THRA1p.(E173G) (blue triangle), and THRA1p.(A263V) (light blue circles) to investigate the transcriptional effect of the variant. (**B**) Transfection with either THRA1(WT) (dark blue squares) alone or with the same amount of THRA1(WT) and THRA1p.(E173G) (blue diamonds) to model the patients’ heterozygous state shows a gain-of-function for the variant. Statistical analysis was performed by a Wilcoxon matched-pairs signed rank test. (**C**) In addition to mimicking the heterozygous state in (**B**), we also investigated the homozygous state. Here, the amount of DNA encoding for either THRA1(WT) (dark blue squares) or THRA1p.(E173G) (light blue diamonds) was double the amount as in (**A**). The gain-of-function effect of THRA1p.(E173G) was over-proportionally enhanced, suggesting a gene-dosage effect. For statistical analysis, a Wilcoxon matched-pairs signed-rank test was performed comparing the two datasets. (**D**) Investigation of the impact of THRA2p.(E173G) on THRA1(WT) revealed a strong antagonistic effect as compared to THRA2(WT) when transfected in a ratio of 1:1. This significant inhibitory effect was also present in comparison to our internal controls of THRA1(WT) + empty vector and of THRA1(WT) + THRA1(WT). Statistical analysis was performed by a non-paired one-way ANOVA comparing individual pairs. (**E**) Co-expression of THRA1(WT) and either THRA2(WT) (red diamonds) or THRA2p.(E173G) (orange triangles) was performed in a concentration-dependent manner, revealing the enhanced antagonism of THRA2p.(E173G), even with an equal amount of THRA1 and THRA2, which was statistically significant according to a two-way ANOVA followed by Dunnett’s post hoc test. THRA2(WT) did not have a strong effect on THRA1(WT) when expressed in equimolar amounts, since it did not differ from THRA1(WT) (dark blue squares). (**F**) THRA1p.(E173G) did not have an effect on the formation of heterodimers with RXRα as titration with different amounts of RXRα did not result in a statistical difference to the WT, which was tested by one-way ANOVA with Kruskal–Wallis test performing individual comparison for each expression ratio. For all panels, cells were lysed after stimulation with ascending concentrations of T3 in (**A–C**,**E**) or with a fixed concentration of 10 nM T3 in (**D**,**F**) for 24 h, and luciferase activity was determined with a luminometer. Values were normalized to the homozygous THRA1/2(WT) state ± SEM for (**A–C**,**E**,**F**), ± SD for (**D**). (**A–C**,**E**) Data resulted from the number of independent experiments provided in the square brackets, each experiment was performed in technical triplicates. (**D**) Data resulted from seven different independent experiments performed in triplicates. ns, not significant; *, *p* ≤ 0.05; **, *p* ≤ 0.01. (**F**) Data resulted from three different independent experiments.

**Figure 3 ijms-22-05338-f003:**
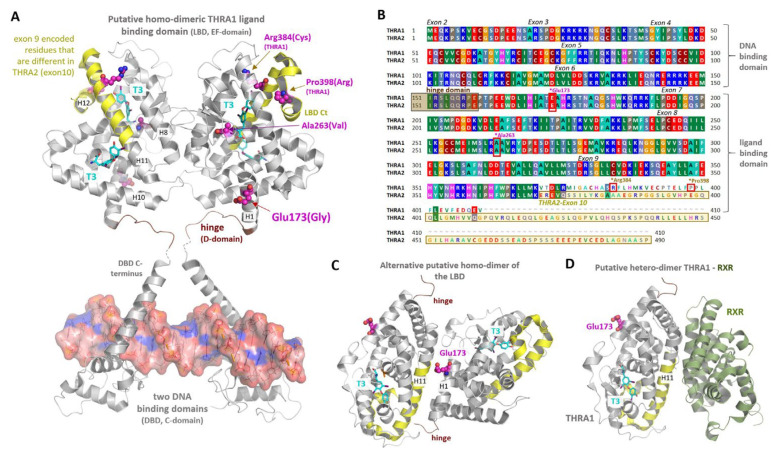
Structural implications for THRA position Glu173. (**A**) Two crystal structures (PDB ID: 4lnw) of a monomeric THRA LBD domain (AA 148–406) including the hinge region (brown) with two TH (magenta sticks) binding pockets were superimposed with the dimeric THRB crystal structure (PDB ID: 3d57) to simulate a putative THRA1 LBD dimer constellation. This arrangement in combination with a crystal structure of a dimeric THR:DNA-binding domain (PDB ID: 3m9e) enables the visualization of structural THR features and the potential impact of the p.(E173G) variant. Glu173 is not involved directly in ligand- nor in DNA-binding. In contrast, known pathogenic THRA variants, such as p.(R384C) (THRA1-specific) or p.(A263V) (THRA1- and THRA2-specific) have an impact on the TH-binding process, either by being located inside the TH-binding pocket, or by being linked to protein parts participating in the TH-binding process (e.g., p.(P398R) in the LBD C-terminus, ligand-pocket entrance). In this putative LBD dimer constellation, protomer contacts are mediated by helix 8 (H8) and the transition between H10 and H11. The structural part that would be encoded partially by exon 10 of THRA2 is marked in yellow. (**B**) The sequence similarities and differences between THRA splicing variants are visualized by the protein sequence alignment. The color-coding distinguishes diverse AA side chain properties, e.g., green for hydrophobic, red for negatively charged, or blue for positively charged. AAs that are encoded by exon 10 (THRA2) are highlighted by a yellow box, which corresponds partially to the yellow backbone-cartoon part in (**A**). Moreover, exon-encoded sequence dimensions are annotated as well as positions of disease-associated variants shown in (**A**). (**C**) An alternative—as compared to (**A**)—homodimeric THRB constellation (PDB ID: 1n46) was used to simulate a second putative dimeric constellation for the THRA LBD, whereby the Glu173 position would be involved in the dimer-interface. (**D**) Suggested by the solved THRB/RXR heterodimer complex (PDB ID: 4zo1) and by assuming similar structural properties for THRA and THRB based on very high sequence identity (~75%), Glu173 is likely not involved in the THR/RXR heterodimer-interface.

**Figure 4 ijms-22-05338-f004:**
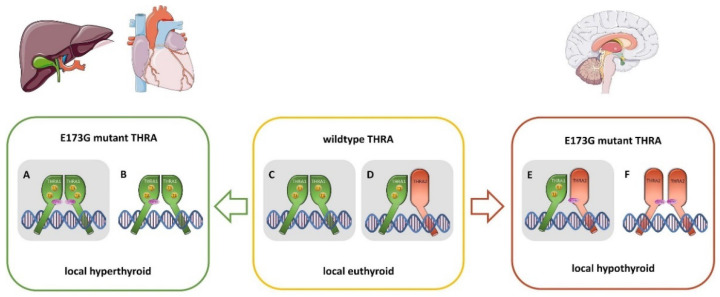
Schematic overview of the putative THRA dimer constellations in patients with the THRAp.(E173G) variant. In the patient, four different isoforms of THRA [THRA1(WT), THRA1p.(E173G), THRA2(WT), THRA2p.(E173G)] exist and may homo- or hetero-dimerize with each other, depending on their relative abundance in respective cells and organs. This may help to explain the symptoms seen in our patients. THRA1p.(E173G) homodimers (**A**) and THRA1(WT):THRA1p.(E173G) heterodimers (**B**), as they are present in peripheral tissues such as cardiomyocytes and hepatocytes, lead to an increased translational activity in these cells, resulting in a local hyperthyroid state as compared to the normally occurring THRA1(WT) homodimer (**C**). At equimolar concentrations, THRA2(WT) has no antagonistic effect on THRA1(WT) (**D**). Tissues that predominantly express THRA2, such as in central neurons, THRA2p.(E173G) has an augmented inhibitory effect on THRA1(WT) (**E**), resulting in a local hypothyroid state in these cells. Other putative heterodimer pairings still have to be investigated, such as THRA2p.(E173G):THRA2p.(E173G) (**F**), or THRA1p.(E173G):THRA2(WT), and Table 1. p.(E173G):THRA2p.(E173G). In this study, we examined the constellations (**A**,**C**,**D**), and (**E**), which are highlighted in grey.

**Table 1 ijms-22-05338-t001:** List of the clinical characteristics seen in the homozygous THRAp.(E173G) mutation carriers that can be associated with resistance to TH. Note: TSH, fT3, and fT4 levels at the time points ① and ② were measured in two different laboratories. Behind the values in brackets, we provide the age-specific normal values of the respective laboratory. The hormone levels at time point ② were measured using the Cobas Elecsys^®^ TSH, T3, T4 laboratory system.

Family Members *			II:4	II:2	I:1
**Variant on the cDNA level**			*THRA* c.[518A > G]; [c.518A > G]	*THRA* c.[518A > G]; [c.518A > G]	*THRA* c.[518A > G]; [c.518A > G]
Variant on the protein level			THRAp.(E173G)	THRAp.(E173G)	THRAp.(E173G)
**Features described to be associated with disease-causing *THRA* variants**					
Anthropometrics	HP:0001513	**Obesity**	yes	yes	yes
		body weight (z-score)	107.9 kg (+3.56 z)	115.6 kg (+3.88 z)	107.0 kg (+2.59 z)
		Body mass index (BMI) (z-score)	44.4 kg/m^2^ (+4.12 z)	44.3 kg/m^2^ (+4.11 z)	37.9 kg/m^2^ (+3.09 z)
	HP:0004322	**Short stature**	no	no	no
		Body height (z-score)	156 cm (–1.89 z) **	161.6 cm (–1.02 z)	168.0 cm (–1.85 z)
	HP:0000256	**Macrocephaly**	no	no	yes
		Head circumference (z-score)	55.5 cm (+0.18 z)	56.4 cm (+0.86 z)	60.5 cm (+2.31 z)
Dysmorphic features	HP:0005280	**Flat nasal bridge**	yes	yes	no
	HP:0000283	**Broad facies**	yes	yes	yes
	HP:0000158	**Macroglossia**	yes	no	no
	HP:0000280	**Coarse facial features**	yes	yes	no
	HP:0010609	**Skin tags**	no	no	no
Neurological function	HP:0001263	**Global developmental** **delay**	yes	yes	no
	HP:0002283	**Brain atrophy on MRI**	yes (mild)	yes	no MRI done
	HP:0001260	**Dysarthria**	yes	yes	no
	HP:0000750	**Delayed speech** **development**	yes	yes	no
	HP:0004302	**Motor dyspraxia**	yes	no	no
	HP:0002375	**Slow initiation** of **movement**	yes	yes	no
	HP:0002066	**Gait ataxia**	yes	no	no
	HP:0002075	**Dysdiadochokinesis**	yes	no	no
Cognitive function	HP:0001256	**Intellectual disability (mild)**	yes	yes	no
		IQ assessment	IQ = 60 (at 7 years)	IQ = 68 (at 8 years)	not tested
	HP:0000708	**Behavioral abnormalities**	yes	yes	no
Skeletal development	HP:0000684	**Delayed eruption of teeth**	yes	yes	no
Gastro-intestinal function	HP:0002019	**Constipation**	yes	yes	yes
Endocrine system	HP:0031508	**Abnormal thyroid** **hormone levels**	no	no	no
		free T3 (age-specific REF)	① 4.84 ng/L (4.70–7.20) ② 3.37 ng/L (3.00–4.90)	① 4.45 ng/L (3.50–6.50) ② 3.24 ng/L (2.00–4.40)	① 6.16 ng/L (3.50–6.50) ② 4.11 ng/L (2.00–4.40)
		free T4 (age-specific REF)	① 15.43 ng/L (10.70- 18.70) ② 12.00 ng/L (9.00–15.00)	① 13.62 ng/L (10.70–18.70) ② 11.40 ng/L (9.30–11.00)	① 13.07 ng/L (10.70–18.70) ② 11.30 ng/L (9.30–17.00)
		TSH (age-specific REF)	① 1.48 mU/mL (0.51–4.94) ② 1.39 mU/mL (0.91–2.18)	① 1.14 mU/mL (0.55–4.78) ② 2.17 mU/mL (0.27–4.20)	② 3.14 mU/mL (0.27–4.20)
		fT3/fT4	① 0.31 ② 0.28	① 0.33 ② 0.28	① 0.47 ② 0.36
Blood	HP:0001903	**Anemia**	no	no	no
		Hemoglobin (g/dL)	13.2 (12.0–15.4)	11.0 (12.0–165.4)	16.2 (13.5–17.0)
		Red blood count (/pl)	5.0 (4.1–5.1)	4.6 (4.1–5.1)	5.2 (4.5–5.3)
Cardio- vascular system	HP:0001688	**Bradycardia**	no	no	no
		Heart rate (/min)	104 (tachycardia)	80	80
	HP:0002615	**Arterial hypotension**	no	no	no
		Blood pressure (mmHg)	130/89	120/83	135/85

* Numbers refer to the family pedigree in Figure 1C. ** The body height was corrected for scoliosis. ① & ②: hormone measurements at two different time points (ca. 2 years apart), REF: reference values

## Data Availability

For patient privacy reasons, genomic data are not allowed by the IRB of Charité to be shared or uploaded to a public repository. The raw data of the non-genetic experiments will be made available to investigators upon reasonable request. The genetic variant has been submitted to the ClinVar database. It is accessible there under the following URL: https://www.ncbi.nlm.nih.gov/clinvar/variation/992628/.

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
