# Peer review of "A New Mechanism in THRA Resistance: The First Disease-Associated Variant Leading to an Increased Inhibitory Function of THRA2"

_ijms, 2021, doi:10.3390/ijms22105338_

Round 1
Reviewer 1 Report
Because the manuscript proposes a new mechanism for the effects of a THR-related mutation, the burden of proof required is substantial. However, the conclusions of the manuscript are based on differences shown in the biological analyses in Figure 2, and specifically in panels D and E. However, these differences are very minor and not highly inconsistent between the two panels. Moreover, the technical description of the methods on which these critical results are obtained is very deficient, raising questions about the validity of the results.
Main comments:
- The choice of human placenta choriocarcinoma (JEG3) cells is not explained, and the tranfectability of these cells is not well documented in the manuscript. Also, the expression levels of endogenous THRs in these cells are not shown.
- It is unclear how the quantity of plasmids to be transfected was selected. This parameter can have an impact of the results. Was a range of concentrations tested first? In the same sense, transfecting the “double dose” to “model homozygosity” seems rather meaningless.
- Because the differences in the aforementioned panels are very minor, they may have been influenced by differences in transfection efficiency. It is not clear how this parameter was checked in the various experiments shown in Figure 2.
- The legend of Figure 2 states that “data resulted from three to seven different independent experiments performed in triplicates”. Several questions can be raised here: First, Why were some experiments performed 3 times and others 7 times? Second, in panel D, several groups have more than 3x7=21 data points (e.g. about 30 for the black group), which contradicts the aforementioned statement. Third, what do the error bars show? The variance of the independent experiments or the variance of all technical and biological replicates together? This obviously influences the respective p-values.
- To give another example, in panel D, at a T3 concentration of 10 nM, the mean of the THRA1+THRA2 (1:1) group (red) appears slightly lower than the THRA1+THRA1 (1:1) group (blue). However, in panel E, at the same T3 concentration (10-9M = 10 nM), the mean of the red group is actually higher. Even though no difference is claimed between these two groups, this discrepancy raises doubts about the robustness of the data.
Other comments :
- Introduction: In THRB resistance, patients do not always have hyperthyroid symptoms, they can also be paucisymptomatic.
- Introduction: Biological findings in THRA resistance can include normal TSH, low or low-normal T4, and high or high-normal T3.
- Introduction: In the symptoms of THRA resistance, normocytic anemia and bradycardia can also be mentioned.
Abbreviations should be more systematically spelled out (e.g., BW, ADHD, etc.). MDPI journals generally foresee a manuscript section listing all abbreviations in the text.
Author Response
Reviewer #1: Because the manuscript proposes a new mechanism for the effects of a THR-related mutation, the burden of proof required is substantial. However, the conclusions of the manuscript are based on differences shown in the biological analyses in Figure 2, and specifically in panels D and E. However, these differences are very minor and not highly inconsistent between the two panels. Moreover, the technical description of the methods on which these critical results are obtained is very deficient, raising questions about the validity of the results.
Main comments:
[1] The choice of human placenta choriocarcinoma (JEG3) cells is not explained, and the tranfectability of these cells is not well documented in the manuscript. Also, the expression levels of endogenous THRs in these cells are not shown.
Answer: We appreciate the reviewer’s comment. JEG3 is a classical cell line, which has been used in almost all publications concerning THRA mutations, such as in the publication that first described a THRA mutation [1] and in many following studies [2–5]. In these studies the transfectability of these cell lines was proven. Hence we decided to utilize these cells in order to make our results comparable to previously published data.
The basic THR expression levels in JEG3 are not known, however, since also mock-transfected cells somewhat respond to T3 (as now shown in Figure S2), we suspect the JEG3 cells to express thyroid hormone receptors in small amounts. This assumption has been made previously by other authors as well [6]. As thyroid hormone action seems to be essential for cell survival, there are no cell lines known that entirely lack any thyroid hormone receptor.
We have now discussed the subject of choice of the cell line in section 2.3 and provide a graph in the Supplemental Materials (Figure S2) depicting the basal reactivity of JEG3 cells towards thyroid hormone.
[2] It is unclear how the quantity of plasmids to be transfected was selected. This parameter can have an impact of the results. Was a range of concentrations tested first? In the same sense, transfecting the “double dose” to “model homozygosity” seems rather meaningless.
Answer: We apologize for not making our experimental approach entirely clear. Lipofectamine® 2000 was published to be effective as a transfection agent for JEG3 cells (e.g. in [7]). Secondly, Lipofectamine® 2000 is specifically recommended by the manufacturer for this cell line.
Before we started the experiments, we performed several transfection optimization experiments, following the recommendations of the manufacturer (ThermoFisher [8]). For that, a constant amount of DNA (2.5 µg per triplicate) was distributed in equimolar amounts between two plasmids, one containing the reporter and the other the THRA gene. We tested different cell numbers and amounts of transfection reagent according to the manufacturer’s protocol. Based on the results obtained from these experiments, we decided upon 20,000 cells per well and a 1:1 (v/v) ratio of DNA:Lipofectamine® 2000. The setup was optimized for transfection with THRA1 + empty vector + reporter, which would represent the expression of a single (wildtype versus mutant) allele. The total amount of transfected DNA was kept constant between the experiments. In order to model the heterozygous situation in the patients for the mutation, we added respective mixtures of the THRA1(wt) + THRA1(mut) plasmids + reporter, again with a constant amount of total DNA. We are aware that in vitro systems are always artificial and can only provide an idea of the physiological mechanisms. We have discussed this caveat in the manuscript in section 4.4.
[3] Because the differences in the aforementioned panels are very minor, they may have been influenced by differences in transfection efficiency. It is not clear how this parameter was checked in the various experiments shown in Figure 2.
Answer: We thank the reviewer for this comment. Even though the effects are considered to be minor by the reviewer, the data were obtained from at least n=3 independent experiments.
Variable transfection efficiency is a general problem of the method. However, such variations of transfection efficiency would occur in the test sample as well as in the control sample. Therefore, if there was a difference between test and control samples of sufficient size, it should still become statistically significant after sufficient repetitions of the experiment.
Unfortunately, most methods that control for transgene expression levels in transfected cells such as Western blot are semi-quantitative as well. Additionally, no reliable antibodies for THRA are presently available, let alone antibodies that distinguish between the splicing isoforms or the presence / absence of a missense mutation.
[4] The legend of Figure 2 states that “data resulted from three to seven different independent experiments performed in triplicates”. Several questions can be raised here: First, why were some experiments performed 3 times and others 7 times? Second, in panel D, several groups have more than 3x7=21 data points (e.g. about 30 for the black group), which contradicts the aforementioned statement. Third, what do the error bars show? The variance of the independent experiments or the variance of all technical and biological replicates together? This obviously influences the respective p-values.
Answer: We thank the reviewer for this very important comment. To answer the first part of the question: the number of experiments cited in the original legend referred to all panels, so it is true that some experiments, such as the concentration-response curves in panel E were only performed three to five times, whereas experiments of panel D were performed seven times.
In panel E we provide a dose response curve (with 3 experiments per data point), while in panel D we have kept the T3 concentration constant at 10 nM to demonstrate that THRA2p.(E173G) has a much stronger inhibitory function than THRA2(WT). The T3 concentration of 10 nM is considered in the literature to be more physiological and other investigators have used this concentration as well, see Moran et al. 2014 [2].
In the first submission of the manuscript we have included the data points at 10 nM from the 1:1 ratio of the panel E experiment into the respective portions of the panel D graphs. However, as this has obviously led to some confusion about the n-numbers, we now removed these data points so that all graphs on panel D derive from n=7 repetitions. This alteration does change the significance levels between THRA1(WT):THRA2p.(E173G) ratio of 1:1 and THRA1(WT):THRA2(WT) ratio of 1:5 to a p-value below 0.05 of panel D. We now provide the exact n-numbers of repetitions for each experiment directly on the panels of Figure 2.
To answer the third part of the question: As stated in the figure legend, all measurements on the panels except D were depicted with their standard error (SEM). For panel D we decided to use a scatter plot, which is best depicted with its standard deviation (SD). In contrast to other methods such as qPCR, the technical replicates are not simply repetitive measurements of the same well. The assays were designed in a way that we created replicates that originated from the same cell flask, were seeded into three individual wells and transfected individually with a transfection master mix for each combination. Each well was then measured individually after cell lysis. For independent experiments, these assays were repeated with cells from another passage and with a new transfection mix. This results in the “technical replicates”, which are cell passage- and transfection mix-dependent triplicates and the “biological replicates”, which are independent experiments performed in consecutive weeks. The data points of all experiments were evaluated together. We now describe this in more detail in the methods section.
[5] To give another example, in panel D, at a T3 concentration of 10 nM, the mean of the THRA1 + THRA2 (1:1) group (red) appears slightly lower than the THRA1 + THRA1 (1:1) group (blue). However, in panel E, at the same T3 concentration (10-9M = 10 nM), the mean of the red group is actually higher. Even though no difference is claimed between these two groups, this discrepancy raises doubts about the robustness of the data.
Answer: We agree with the reviewer that the differences mentioned in the two groups are not significant, neither on panel D nor on panel E. Due to large SEM and relatively low number of replicates, panel E shows a slightly higher average activity at 10 nM. Exactly for this reason, in order to make the data more robust and increase statistical power, we have repeated the experiment 7 times, which are now depicted in panel D. As both statistical comparisons in the panel E and D experiments come to the same conclusions, we cannot see any fault with the robustness of our data.
[6] Introduction: In THRB resistance, patients do not always have hyperthyroid symptoms, they can also be paucisymptomatic.
Answer: We have mentioned this now in the introduction.
[7] Introduction: Biological findings in THRA resistance can include normal TSH, low or low-normal T4, and high or high-normal T3.
Answer: We have mentioned this now in the introduction.
[8] Introduction: In the symptoms of THRA resistance, normocytic anemia and bradycardia can also be mentioned.
Answer: We have mentioned this now in the introduction.
[9] Abbreviations should be more systematically spelled out (e.g., BW, ADHD, etc.). MDPI journals generally foresee a manuscript section listing all abbreviations in the text.
Answer: We have now spelled out all abbreviations at first mentioning and added a list of all abbreviations and acronyms occurring in the text. Rarely occurring abbreviations have been removed entirely.
Reviewer 2 Report
Excellent manuscript. Very well written, presents clearly the results and the conclusion. Novel and very interesting. My only minor comment would be for the authors to include the assay types by which TSH, FT4, FT3 were measured as there are also different normal ranges reported. Also, if there are pictures of the I1 father it would be interesting to include.Author Response
Reviewer #2
[1] Excellent manuscript. Very well written, presents clearly the results and the conclusion. Novel and very interesting. My only minor comment would be for the authors to include the assay types by which TSH, FT4, FT3 were measured as there are also different normal ranges reported. Also, if there are pictures of the I1 father it would be interesting to include.
Answer: We thank the reviewer for his/her encouraging comments. Unfortunately we were not permitted to include a photographic image of the father, because he did not provide consent.
We have now included into the table legend the type of assay used for the second time point of hormone measurement. The first measurement was done more than 2 years ago at a clinical laboratory in the patients’ home town in Israel and we now provide the age and laboratory specific normal values for these results as well, as obtained from the report sheet. We were unable to exactly find out what kind of commercial assay was used in this lab at the specific time point. We now added to the table legend the following sentence. “nota bene: TSH, fT3, and fT4 levels at the time point â‘ and â‘¡ were measured in two different laboratories. Behind the values in brackets we provide the laboratory- and age-specific normal values. The hormone levels at time point â‘¡ were measured using the Cobas Elecsys® TSH, T3, T4 laboratory system.”
Round 2
Reviewer 1 Report
I regret that I do not find the authors’ responses reassuring regarding the technical reliability and robustness of the experiments. To give some examples, regarding points 3 and 4: (i) The authors acknowledge that transfection efficiency was not formally controlled for. Therefore, the minor differences observed may very well reflect chance findings. (ii) Now that some of the data points from the black group in panel D have been removed (fewer data points), a p-value has become significant, whereas before (with more data points) it was not. This also strongly indicates chance findings, and I don’t think that such data can be taken at face value. (iii) The authors state that “In contrast to other methods such as qPCR, the technical replicates are not simply repetitive measurements of the same well”. This statement also makes me very skeptical in general. I don’t think that in qPCR “technical replicates” refer to repeat measurements of the same well; it refers to duplicate or triplicate wells. (iv) According to their descriptions, the authors analyzed data points from biological and technical replicates together (“The data points of all experiments were evaluated together”). This is inappropriate, because it minimizes the variance and thus increases the chance of statistically significant results (essentially, the technical replicates in each experiment are treated as independent biological replicates). The correct practice is to use the technical replicates in each independent experiment to calculate a mean, and then compare the means of independent experiments (biological replicates) among groups.
As mentioned in the initial review round, the burden of proof required for a new pathogenetic mechanism is high, and must be supported by credible data. This burden of proof has not been met in this study.